# Electrochemical Removal of Nitrogen Compounds from a Simulated Saline Wastewater

**DOI:** 10.3390/molecules28031306

**Published:** 2023-01-30

**Authors:** Pasquale Iovino, Angelo Fenti, Simona Galoppo, Mohammad Saleh Najafinejad, Simeone Chianese, Dino Musmarra

**Affiliations:** 1Department of Environmental, Biological and Pharmaceutical Sciences and Technologies, University of Campania “Luigi Vanvitelli”, Via Vivaldi 43, 81100 Caserta, Italy; 2Department of Engineering, University of Campania “Luigi Vanvitelli”, Via Roma 29, 81031 Aversa, Italy

**Keywords:** nitrogen compound electrochemical oxidation, nitrogen molar balance, ammonium ion, ammonia, parameter investigation, electrochemical degradation pathway

## Abstract

In the last few years, many industrial sectors have generated and discharged large volumes of saline wastewater into the environment. In the present work, the electrochemical removal of nitrogen compounds from synthetic saline wastewater was investigated through a lab-scale experimental reactor. Experiments were carried out to examine the impacts of the operational parameters, such as electrolyte composition and concentration, applied current intensity, and initial ammoniacal nitrogen concentration, on the total nitrogen removal efficiency. Using NaCl as an electrolyte, the *N_TOT_* removal was higher than Na_2_SO_4_ and NaClO_4_; however, increasing the initial NaCl concentration over 250 mg·L^−1^ resulted in no benefits for the *N_TOT_* removal efficiency. A rise in the current intensity from 0.05 A to 0.15 A resulted in an improvement in *N_TOT_* removal. Nevertheless, a further increase to 0.25 A led to basically no enhancement of the efficiency. A lower initial ammoniacal nitrogen concentration resulted in higher removal efficiency. The highest *N_TOT_* removal (about 75%) was achieved after 90 min of treatment operating with a NaCl concentration of 250 mg·L^−1^ at an applied current intensity of 0.15 A and with an initial ammoniacal nitrogen concentration of 13 mg·L^−1^. The nitrogen degradation mechanism proposed assumes a series–parallel reaction system, with a first step in which NH_4_^+^ is in equilibrium with NH_3_. Moreover, the nitrogen molar balance showed that the main product of nitrogen oxidation was N_2_, but NO_3_^−^ was also detected. Collectively, electrochemical treatment is a promising approach for the removal of nitrogen compounds from impacted saline wastewater.

## 1. Introduction

Over recent years, the rapid growth of industrialization and urbanization has produced large amounts of wastewater. Many industries, including food processing, agricultural, petroleum, and textile dyeing, generate huge volumes of wastewater daily with a high salt content [1,2,3].

Discharging this saline wastewater into the environment can result in several problems, such as the contamination of various resources, the fluctuation of salinity levels of water bodies, and the eutrophication of lakes [4,5]. In general, saline wastewater is characterized by salt in the range of 1–3.5% *w*/*w*; conversely, wastewater with higher salt content is defined as hypersaline [6,7]. Nevertheless, the salinity of the wastewater is affected by the different industrial processes involved. For instance, tannery industries may produce wastewater with a salinity of even 8%, whereas seafood wastewater can contain a number of salts in the range of 2–4% [8].

The salinity levels of the wastewater are generally associated with the presence of dissolved inorganic salts, such as NaCl, Na_2_SO_4_, CaClO_2_, and MgSO_4_ [9,10]. Among others, ammonium chloride (NH_4_Cl) is a low-cost nitrogen inorganic salt widely detected in various industrial wastewaters [11]. NH_4_Cl constitutes one of the major sources of ammoniacal nitrogen (NH_4_^+^-NH_3_); consequently, its uncontrolled use could lead to serious challenges. During heating processes, NH_4_Cl may release NH_3_ and HCl; in the chlorination of drinking water, this can lead to the formation of disinfection byproducts (DBPs), such as haloacetamides [12,13]. In addition, NH_4_Cl, whether as gas, solid, or liquid, is a corrosive agent [14]. In an aqueous solution, NH_4_Cl is dissociated in ammonium ions (NH_4_^+^) and chloride (Cl^−^), and, as well known, NH_4_^+^ is under pH-controlled equilibrium with ammonia (NH_3_) (pK_a_ = 9.25) [15,16,17].

Above pH 10, the prevalent species available in solution is NH_3_, whereas, under pH 7, NH_4_^+^ dominates. In the pH range between 7 and 10, both species are in equilibrium. The neutral molecule of ammonia can diffuse across the epithelial membranes of an organism, causing heavy damage such as asphyxiation, inhibition of the Krebs cycle, and functional decline of the liver and kidney [18,19,20]. Due to the easy interchangeability between NH_4_^+^ and NH_3_, and since ammonium is the predominant form (>90%) over ammonia (NH_3_) in the majority of water systems at pH < 8.2 and temperature < 28 °C [21], NH_4_^+^-NH_3_ removal from impacted water has become increasingly prominent for the scientific community. Moreover, NH_4_^+^-NH_3_ may form different dangerous compounds in solution, such as nitrite (NO_2_^−^) and nitrate (NO_3_^−^), resulting in further contamination of water bodies [22,23].

In the last few years, many methods have been employed for the removal of NH_4_^+^-NH_3_ from both wastewater and saline wastewater, including adsorption [24,25], reverse osmosis [26], ultra and nanofiltration [27], assimilating biosystems [8], biological methods [28], and microbial fuel cells (MFCs) [29]. Although these techniques have shown sustainability [30,31] and decent ammoniacal nitrogen removal efficiency, there are limitations related to each of them. Adsorption techniques generate concentrated waste streams, resulting in a simple transfer of the contaminant to a different environment domain [32,33]. Reverse osmosis and filtration systems require high energy and costs [34], while biological treatments entail complex operational conditions and high risk during subsequent processes [20]. Compared to the other techniques, electrochemical oxidation (EO) processes have been gaining more attention in recent times [35].

EO is a technique belonging to the class of advanced oxidation processes (AOPs), which have proven to be very effective and reliable for removing emerging pollutants and NH_4_^+^-NH_3_ from water [36,37,38,39,40]. The efficiency of the electrochemical treatments is influenced by several factors, including initial NH_4_^+^-NH_3_ concentration, applied current intensity, pH, concentration and composition of the electrolyte, and type of anode material used [35,41]. Among these factors, the composition of the electrolyte, the applied current density, and the type of anode material used over the treatment are the most significant parameters impacting both efficiency of the process and the overall costs [40].

In general, the electrochemical removal of ammoniacal nitrogen involves two mechanisms, called direct and indirect oxidation [35]. During direct oxidation or anodic oxidation, the contaminant is absorbed on the anode surface, thus favoring a direct electron transfer between the surface and the pollutant molecules. Conversely, indirect oxidation leverages the in situ generation at the anode surface of strong oxidant species, such as hydroxyl radicals (^•^OH) and active chlorine species, which are capable of oxidizing the ammonium present in the solution [11,42]. Generally, in EO processes, it is necessary to add an amount of salt, such as NaCl, into the solution to enhance the conductivity of the system and, in parallel, trigger the in situ generation of active chlorine species. However, since saline wastewater is naturally rich in chloride ions [43,44], the electrochemical production of active chlorine species, including Cl_2_, HOCl, and OCl^−^, is clear, which means that no further addition of salt is necessary. As reported in the literature, EO successfully removes NH_4_^+^-NH_3_ from saline water systems, but several aspects need to be further discussed. Wilk et al. [45] investigated the ammonium nitrogen EO of organic compounds in landfill leachates, characterized by a high salinity of 2690 ± 70 mg Cl^−^·L^−1^. After 8 h of treatment, about 60% of ammoniacal nitrogen was removed operating under the best operative conditions, but the main limitation that occurred was the high energy consumed during the treatment. Dìaz et al. [46] examined an EO treatment of ammoniacal nitrogen from an aquaculture saline water system. Although the removal process has been effective, some drawbacks have been detected, including the formation of oxidation byproducts, i.e., trihalomethanes. Sun et al. [11] reported on the electrochemical chlorine-mediated NH_4_^+^-NH_3_ removal from saline wastewater, in which they achieved about 98% of NH_4_^+^-NH_3_ oxidation at 1.00 mA·cm^−2^ after 2 h of treatment. The formation of NO_3_^−^ and nitrite NO_2_^−^ was also monitored to examine the pathway of NH_4_^+^ during its oxidation.

In light of the gaps reported above, in this study, we examine the EO of ammoniacal nitrogen in simulated saline NH_4_Cl wastewater using bored-doped diamond (BDD) electrodes. BDD electrodes, belonging to the class of nonactive electrodes, in contrast to transition-metal oxide (TMO) active anodes [47,48], are widely recognized as a very stable anodic material with a higher production of ^•^OH and, thus, with a high overpotential of O_2_, implying high efficiency for the removal of ammonium from impacted water [45,49,50].

With this work, we aim to contribute to the exploration of nitrogen removal via electrochemical oxidation, a field not deeply explored in contrast to the removal of other compounds and still debated. The effects of electrolyte composition, chloride concentration, and current intensity on removal efficiency are controversial. For example, Kapalka et al. [51] indicated that the direct EO pathway could oxidize ammonia on the BDD anode surface. This result was confirmed by Zollig et al. [52]. Conversely, Candido et al. [16] reported a poor contribution of direct EO on the removal of ammoniacal nitrogen, likely due to the possible formation of incompletely oxidized adsorbed nitrogen species and ^•^OH on the anode surface, resulting in shielding from direct oxidation. Mandal et al. [35] reported that the ammonia oxidation increased when the initial chloride concentration increased from 300 to 1500 mg·L^−1^; however, in the range from 300 to 900 mg·L^−1^ the ammonia removal percentage did not change significantly. In contrast, Li et al. [53] showed a linear correlation between the ammonia removal efficiency and the initial chloride concentration across the investigated range, confirming the results of other studies [54,55]. Shih et al. [56] an appreciable impact of the applied current intensity on the EO of ammoniacal nitrogen; a higher current intensity led to higher removal of the contaminant. This trend was confirmed by Zhang et al. [55], but contradicted other previous studies where a decrease in removal efficiency was found since a higher applied current intensity implies an increase in undesired side reactions, such as oxygen evolution and generation of byproducts [38,57]. Therefore, by investigating the parameters mentioned above, the authors would like to contribute to improving the understanding of nitrogen removal via electrochemical oxidation.

Moreover, the manuscript proposes an integrated approach for assessing the effectiveness of the electrochemical nitrogen removal, i.e., if the nitrogen was oxidized to N_2_, representing the optimal result, or to other species such as NO_3_^−^. Therefore, byproduct formation was investigated and monitored, and a nitrogen balance was performed, which required the definition of a degradation mechanism and the assessment of the volumetric mass transfer coefficient. According to the best of the authors’ knowledge, this is the first time this kind of integrated approach has been proposed for the investigation of ammoniacal nitrogen removal from wastewater.

The impact on the NH_4_^+^-NH_3_ EO process by different types of electrolytes, naturally occurring in saline wastewater, is evaluated. Moreover, to identify the optimal operative conditions, the effect of other parameters is investigated. The mechanism which governs the degradation is proposed, and the fate of nitrogen in the various phases along the treatment is assessed. The latter aspect was scarcely investigated in prior studies.

## 2. Results and Discussion

### 2.1. Electrolyte Composition Impact on the Electroremoval of Ammonium Chloride

Saline wastewater is typically rich in various salts with different compositions, including salts of chloride, sulfate, and nitrate [58,59,60]. As known, the electrolyte composition strongly influences EO processes. To simulate the presence of electrolytes generally present in saline wastewater, NaCl, Na_2_SO_4_, and NaClO_4_ were used to study their effect on the electroremoval of ammonium chloride. The results reported in Figure 1 clearly indicate that, when using NaCl as an electrolyte, the *N_TOT_* removal was higher than when using Na_2_SO_4_ and NaClO_4_, achieving about 75% of *N_TOT_* removal after 150 min of treatment. In particular, *N_TOT_* removal rapidly increased until 60 min of treatment; after that, it was constant. This trend is consistent with previous electrochemical oxidation investigations [61] and can be explained by considering that, over the time of treatment, the reactive chlorine species present in the solution can lead to the formation of undesired byproducts, such as chlorate and perchlorate, among others, which may hinder further oxidation of the contaminant at the BDD anode [62,63,64]. Consequently, after 60 min of treatment, a constant *N_TOT_* removal evolution occurred.

Furthermore, it is worth noting that, when Na_2_SO_4_ and NaClO_4_ were used, no *N_TOT_* removal was basically achieved at the end of the process. The high removal efficiency gained in the presence of NaCl proves that around 75% of *N_TOT_*, initially present in the solution as NH_4_Cl, was removed after 150 min. As reported in a previous study by Mandal et al. [35], the presence of chloride in the solution promotes a very effective electroremoval process of NH_4_^+^-NH_3_.

The EO of NH_4_^+^-NH_3_ is typically mediated by two mechanisms, depending on the presence or absence of chloride [16]. When chloride is not in solution, i.e., for the experimental runs with Na_2_SO_4_ and NaClO_4_ as electrolytes, the removal of NH_4_^+^-NH_3_ may occur through direct oxidation on the anode surface, resulting in the formation of gaseous nitrogen as the final main product (Equations (1) and (2)) [16,42,51].
2 NH_4_^+^ → N_2_ + 8 H^+^ + 8 e^−^.(1)
2 NH_3_ → N_2_ + 6H^+^ + 6 e^−^.(2)

The experimental results prove that the direct mechanism seems ineffective since no *N_TOT_* removal was achieved using Na_2_SO_4_ and NaClO_4_ at the end of the respective treatments. Candido et al. [16] reported that the direct electrochemical oxidation of ammonium nitrogen can be affected by the formation of incompletely oxidized nitrogen species and ^•^OH generated during the process on the anode surface, resulting in a reduction in the oxidation efficiency. This achievement was also reported in several studies, in which it was highlighted that one of the main limitations of ammoniacal nitrogen oxidation is represented by the competition between the adsorption of ammoniacal species and ^•^OH on the anode surface, causing a blocking effect on its active zones [65,66]. The results showed in our work are consistent with previous studies [62], where it is reported that nitrogen compounds are known to be the main species poisoning (deactivating) the anode surface during ammoniacal nitrogen oxidation.

Nevertheless, the results contrast with the findings reported by Bagastyo et al. [50], where the presence of Na_2_SO_4_ enhanced the ammoniacal nitrogen electroremoval.

Conversely, when NaCl is used, the chloride present in the solution can trigger the generation of chlorine-active species, according to the following reactions [38,67,68]:2 Cl^−^ → Cl_2_ + 2 e^−^.(3)
Cl_2_ + H_2_O → HClO + H^+^ + Cl^−^.(4)
HOCl → OCl^−^ + H^+^.(5)

Since, under all of the experimental conditions investigated, the pH of the solution was under 6, HClO was the main chlorine active species involved in the EO of NH_4_^+^-NH_3_ [69]. The hypochlorous acid formed can indirectly oxidize the NH_4_^+^-NH_3_ in a reaction zone near the anode surface [62] into nitrogen gas due to its high oxidative potentials in the so-called indirect EO mechanism [41,70].
2 NH_4_^+^ + 3 HOCl → N_2_ + 3 H_2_O + 5 H^+^ + 3 Cl^−^.(6)
2 NH_3_ +3 HOCl → N_2_ + 3 H_2_O + 3 HCl.(7)

These findings agree with Dìaz et al. [46], who successfully removed ammoniacal nitrogen from impacted water by indirect EO through in situ electrogenerated HClO [46,71]. The reactions, shown in Equations (6) and (7), may represent the main mechanisms of NH_4_^+^-NH_3_ oxidation in our system. Pèrez et al. [72] also reported that the main NH_4_^+^-NH_3_ oxidation product obtained during the electrochemical treatment was N_2,gas_, with a percentage around 80%. Nevertheless, as stated in well-known studies on breakpoint chlorination, the EO of NH_4_^+^-NH_3_, in the presence of HClO, can also result in NO_3_^−^ formation [73].
NH_4_^+^ + 4 HOCl → NO_3_^−^ + H_2_O + 6 H^+^ + 4 Cl^−^.(8)
NH_3_ + 4 HOCl → NO_3_^−^ + H_2_O + H^+^ + 4 HCl.(9)

It can be considered that implementing chloride during the EO represents a suitable process to minimize NH_4_^+^-NH_3_ content in impacted water [74].

### 2.2. Impacts of Varying Process Parameters on the Electroremoval of Ammonium Chloride

#### 2.2.1. Effect of Chloride Concentration

Since the findings shown above indicate that the electroremoval of NH_4_^+^-NH_3_ strongly depends on the presence of chloride, investigations of the effect of NaCl concentration on the EO of NH_4_^+^-NH_3_ were carried out by varying the salt concentration in the range 100–750 mg·L^−1^ (experimental run 2 in Table 1). The results are displayed in Figure 2.

As can be seen, after 150 min of treatment, 21.8% of *N_TOT_* removal was achieved operating at the lowest NaCl concentration of 100 mg·L^−1^. Upon increasing the initial NaCl concentration to 250 mg·L^−1^, marked improvements in terms of *N_TOT_* removal (74.5%) were achieved at the end of the process, but further addition of NaCl (500 and 750 mg·L^−1^) resulted in no benefits in terms of the *N_TOT_* removal efficiency. This outcome could suggest that a smaller amount of NaCl (250 mg·L^−1^) is required to obtain the same percentage of *N_TOT_* removal. When operating at higher NaCl concentrations, i.e., 500 or 750 mg·L^−1^, the system could need a higher applied current intensity than 0.15 A to generate more active chlorine species. Thus, the reaction reported in Equation (3) may represent the limiting step for the process, acting under high salt concentration conditions and low applied current intensities [53].

#### 2.2.2. Effect of Applied Current Intensity

Figure 3 depicts the effect of the applied current intensity on the *N_TOT_* removal (experimental run 3 in Table 1).

*N_TOT_* removal efficiency of 57.8% was obtained operating with the lowest applied current intensity of 0.05 A after 150 min of treatment, despite the linear increase observed. A rise in the current intensity to 0.15 A resulted in an improvement of *N_TOT_* removal of 74.5% after a treatment time of 90 min; after that, the removal efficiency was constant. A further increase to 0.25 A led to no enhancement of the efficiency, showing a reduction in the removal efficiency at about 72% after 150 min.

Acting at the lower applied current intensity, the removal of ammoniacal nitrogen is slower [75]. However, Piya-areetham et al. [76] reported that, when operating at lower current intensity, the contaminant removal still increased after 360 min of treatment, confirming the trend shown in our work.

Increasing the current intensity from 0.05 A to 0.15 A resulted in an increase in the *N_TOT_* removal rate. This achievement can be explained by considering a faster production of hydroxyl radicals and reactive chlorine species, as stated in several previous studies on electrochemical oxidation processes [77], which favor the oxidation of the organic compounds, i.e., the achievement of higher removal efficiencies. On the other hand, over time of treatment, the reactive chlorine species present in the solution can lead to the formation of undesired byproducts, such as chlorate and perchlorate, among others, which may hinder further oxidation of the contaminant at the BDD anode [62,63,64], justifying the constant removal efficiency after 90 min. Similar trends were shown in several previous studies [75,78].

In theory, increasing the applied current intensity implies a consequent higher production of active chlorine species, speeding up the oxidation reaction and enhancing the process removal efficiencies [61,79]. However, it also means a decrease in both the selectivity and the current efficiency of the system since a higher applied current intensity implies an increase of undesired side reactions, such as oxygen evolution and generation of byproducts (chlorate and perchlorate) [38,57], which reduces the removal efficiency.
6 HClO + 3 H_2_O → 2 ClO_3_^−^ + 4 Cl^−^ + 12 H^+^ + 3/2 O_2_ + 6 e^−^.(10)
ClO_3_^−^ + H_2_O → ClO_4_^−^ + 2H^+^ + 2 e^−^.(11)

#### 2.2.3. Effect of Initial Ammonium Concentration

The initial concentration of ammonium constitutes an important operative parameter, impacting both the removal efficiency and the mechanism of the process [53,80].

Considering the findings reported above, it was decided to examine the effect of the latter factor, in the range of 50–750 mg·L^−1^, on the *N_TOT_* removal efficiency. Figure 4 clearly displays that, upon enhancing the concentration of ammonium initially present in the solution, the *N_TOT_* removal efficiency decreased.

About 64.1%, 20.8%, and 9.7% *N_TOT_* removal was achieved after 180 min of treatment, operating at 50, 250, and 750 mg·L^−1^ initial ammonium concentration, respectively. As stated above, since the indirect EO mechanism mediated by HClO (Equations (6) and (7)) may constitute the main oxidation pathway of NH_4_^+^-NH_3_, the rate at which HClO was produced was the control factor in our process. Therefore, to enhance the initial ammoniacal nitrogen concentration, the system needed a higher amount of NaCl to generate more HClO. Moreover, it is worth noting that the *N_TOT_* removal (%) followed a linear trend over the treatment time for all conditions investigated. This result could suggest that the *N_TOT_* oxidation rates were described by pseudo-zero-order kinetics, indicating that the *N_TOT_* oxidation rate is independent of the initial contaminant concentration. The outcomes agree with previous studies that reported a linear decrease in ammoniacal nitrogen removal over the treatment time [53,81].

### 2.3. Degradation Mechanism and Nitrogen Molar Balance

The nitrogen electro-oxidation pathway proposed in this paper, sketched in Figure 5, was assumed to be a series–parallel reaction system, with a first step in which NH_4_^+^ is in equilibrium with NH_3_. Nitrogen electrochemical oxidation in the presence of Cl^−^ depends on both hydroxyl radicals and active chlorine species and results in the formation of NO_3_^−^ and N_2_ [62,70,73]. In addition, NH_3_ stripping was considered. Yao et al. [75] showed that the mechanism of ammoniacal nitrogen removal depends on both the hydroxyl radical and active chlorine, suggesting that the contaminant could be efficiently oxidized by these oxidants. Several studies have reported that nitrogen gas and nitrate are the main products of electrochemical oxidation of ammoniacal nitrogen [35,62,82].

The results of the nitrogen molar balance are reported in Figure 6. The main product of nitrogen oxidation was N_2_, in agreement with the scientific literature [35]. In particular, N_2_ was one order of magnitude higher than NO_3_^−^. Moreover, at the pH investigated (<7), ammonia stripping was close to zero. This finding is consistent with the scientific literature [42].

## 3. Materials and Methods

### 3.1. Reagents

NH_4_Cl was purchased from Sigma-Aldrich (St. Louis, MO, USA). Sodium chloride, sodium perchlorate, sodium sulfate, and sodium nitrate were used as received from various chemical suppliers. All solutions were prepared with Milli-Q water (18.2 MΩ·cm^−1^ resistivity, 25 °C) from an Elix ^®^ Essential 10 UV water purification system (Merck, Darmstadt, Germany).

### 3.2. Electrochemical Oxidation Experiments

EO experiments were performed in a lab-scale batch reactor with a volume of 0.250 L at 25 °C. The initial NH_4_Cl concentration varied between 50 and 750 mg·L^−1^, resulting in an initial total nitrogen compound (*N_TOT_*) concentration of 13–200 mg·L^−1^. These conditions were implemented to investigate the effect of the initial NH_4_Cl on the electroremoval efficiency and, in parallel, to simulate the typical concentration of saline wastewater partially. The pH of the solution was continuously monitored using a HI 9017 Hanna Instruments pH meter. Under all conditions investigated, the initial pH of the solution resulted under 6.0. As reported above, at this weak acidic value of pH, NH_4_^+^ is the predominant NH_4_^+^-NH_3_ form [16]; however, NH_3_ could also be detected in the system and, therefore, oxidized [21].

The electrochemical cell consisted of two BDD electrode plates (Neocoat, Switzerland), having a size of 100 × 50 mm, with an active area of 50 cm^2^ for each one and a gap of 1 cm. The reactor configuration used in this study is the most employed one at the lab scale for removing ammoniacal nitrogen and other contaminants due to its ease of installation, handling, sampling, and efficiency [53,75,83,84].

A bench-top direct current power supply BPS-305 (Lavolta, London, UK) was connected to the BBD electrodes, enabling it to operate in amperostatic conditions. Figure 7 shows a schematization of the electrochemical reactor.

The effect of the natural presence of different electrolytes in wastewater was simulated by adding NaCl, Na_2_SO_4_, or NaClO_4_ to the NH_4_Cl solution to examine their effect on the degradation process. Further EO tests were carried out to evaluate the effect of the initial salt concentration (100–750 mg·L^−1^), the applied current intensity I (0.05–0.25 A), and the initial NH_4_Cl concentration (50–750 mg·L^−1^). All experimental conditions are reported in Table 1. The experiments were duplicated to determine the reproducibility of the results.

### 3.3. Total Nitrogen Compound (N_TOT_) Removal and Intermediates Analyses

The starting contaminant solution and the aliquots withdrawn at given time intervals during the treatments were analyzed by TOC-L CSH/CSN (Shimadzu, Tokyo, Japan), equipped with a chemiluminescence gas analyzer for *N_TOT_* detection. The analyses were conducted according to the following instrumental conditions: furnace temperature = 720 °C; carrier gas flow = 150.0 mL/min; supply gas pressure = 285.0 kPa.

Accordingly, the EO performances were estimated in terms of *N_TOT_* removal (%) using the following equation:(12)$$N_{TOT}\;removal\;\left(\%\right)=\frac{N_{TOT}\left(t=0\right)-N_{TOT}\left(t\right)}{N_{TOT}\left(t=0\right)}.$$

In addition, NO_3_^−^ measurement was performed using a Lambda 40, spectrometer, with an optical path = 1.00 cm (Perkin Elmer, Waltham, MA, USA).

### 3.4. Degradation Mechanism and Nitrogen Molar Balance

A nitrogen degradation mechanism was proposed, and, for a selected test (experimental run 4 in Table 1), a nitrogen molar balance was carried out by using quantitative analyses of the total nitrogen compound and NO_3_^−^. The molar balance was carried out considering the equilibrium between the ammonium ions and the ammonia, as well as the ammonia stripping due to nitrogen gas. Ammonia stripping was assessed assuming the batch reactor was perfectly stirred and required the assessment of the volumetric mass transfer coefficient *k_L_a*, which resulted from the mass transfer coefficient, *k_L_* (m·s^−1^), and the gas–liquid interfacial area a (m^2^·m^−3^). *k_L_* was calculated according to the following correlation [85]:(13)$$k_{L}=\frac{Sh\;D_{gas,liquid}}{d_{b}},$$
where *Sh* is the Sherwood number, *D_gas,liquid_* is the diffusivity of gas in liquid (m^2^·s^−1^), and *d_b_* is the bubble diameter assumed equal to 1 mm. The Sherwood number, *Sh*, for rising bubbles of gas in liquid as a continuous phase was calculated according to the following correlation [86]:(14)$$Sh=0.95\;Sc^{1/3}\;Re^{1/2},$$
where *Sc* and *Re* are the Schmidt and Reynolds numbers, calculated according to the following equations, respectively [87]:(15)$$Sc=\frac{\mu_{l}}{\rho_{l}\;D_{gas,liquid}},$$
(16)$$Re=\frac{\;d_{b}\;\rho_{l\;}v_{t}\;}{\mu_{l}},$$
where $v_{t}$ (m/s) is the terminal velocity of the bubbles, calculated using the equation reported below [88].
(17)$$v_{t}=\sqrt{\frac{2\sigma_{l\;}}{\rho_{l}d_{b}\;}+\frac{g\;d_{b}}{2}},$$
where $\rho_{l\;}$ (kg·m^−3^), $\rho_{g\;}$ (kg·m^−3^), and ${}_{l\;}$ (N·m^−1^) are the density of the liquid phase, the density of the gas phase, and the surface tension of the liquid phase, respectively, and *g* (m·s^−2^) is the gravitational acceleration.

The gas–liquid interfacial area was calculated according to the following equation [85]:(18)$$a=6\frac{\varepsilon_{g}}{d_{b}},$$
where $\varepsilon_{g}$ is the gas holdup [85].
(19)$$\frac{\varepsilon_{g}}{1-\varepsilon_{g}}=\frac{U_{g}}{0.3+2\;U_{g}},$$
where $U_{g}$ (cm·s^−1^) is the superficial gas velocity, $\rho_{g}$ (kg·m^−3^) is the density of the gas phase, σ is the surface tension (mN·m^−1^), and P the operative pressure in the reactor (MPa). The superficial gas velocity $U_{g}$ was assumed equal to the terminal velocity of the bubble $v_{t}.$ The nitrogen molar balance is described by the following equation system:(20)$$N_{TOT}\left(t\right)=NH_{4}^{+}\left(t\right)+NO_{3}^{-}\left(t\right),$$
(21)$$NH_{4}^{+}\left(t\right)+H_{2}O\left(t\right)↔NH_{3,\;liq}\left(t\right)+H^{+}(t),$$
(22)$$NH_{3,gas}^{i}\left(t\right)=K_{H,NH_{3}}*NH_{3,\;liq}\left(t\right),$$
(23)$$NH_{3,gas}\left(t\right)=k_{L}a*\left[NH_{3,gas}^{i}\left(t\right)-NH_{3,gas}^{atm}\right],$$
(24)$$NH_{4}^{+}\left(t\right)+NO_{3}^{-}\left(t\right)+NH_{3,\;liq}\left(t\right)=NH_{3,gas}\left(t\right)+N_{2,gas}\left(t\right),$$
where $K_{H,NH_{3}}$ is Henry’s law constant of ammonia.

Ammonia transfer from the liquid to the atmosphere was assessed using the double-film model, in which the resistance to ammonia transfer on the liquid side was considered equal to zero and considering the content of ammonia in the atmosphere equal to zero ($NH_{3,gas}^{atm}=0$ mmol·m^−3^). The *N*_2,*gas*_ molar stream flow rate was calculated from the molar balance [72].

The values of the parameters used for the assessment of the volumetric mass transfer coefficient and the nitrogen balance are reported in Table 2.

## 4. Conclusions

Removal of nitrogen species by electrochemical oxidation is suitable thanks to the combination of hydroxyl radicals and strong oxidants. Experimental results highlight that optimizing the operative conditions is a significant step for electro-oxidation, as the increase in NaCl concentration at a value higher than 250 mg·L^−1^ resulted ineffective. The increase of the current intensity at a value higher than 0.15 A showed that no benefits could be achieved. On the other hand, a lower nitrogen compound concentration resulted in higher removal efficiency. The highest *N_TOT_* removal (about 75%) was achieved after 90 min of treatment operating with a NaCl concentration of 250 mg·L^−1^ at an applied current intensity of 0.15 A and with an initial ammoniacal nitrogen concentration of 13 mg·L^−1^.

The electrochemical degradation mechanism of nitrogen compounds can be assumed to be a series–parallel reaction system with a first step in which ammonium ions and ammonia are in equilibrium. Then, nitrogen oxidation occurs, forming NO_3_^−^ and N_2_; however, the main product of nitrogen oxidation is N_2_, while ammonia stripping is about zero.

The effectiveness of nitrogen species removal in terms of N_2_ formation as the main compound by electrochemical oxidation was highlighted; however, other investigations are required to optimize the process, e.g., in terms of applied current intensity and NaCl concentration. Moreover, an assessment of the cost-effectiveness of the process is required.

## Figures and Tables

**Figure 1 molecules-28-01306-f001:**
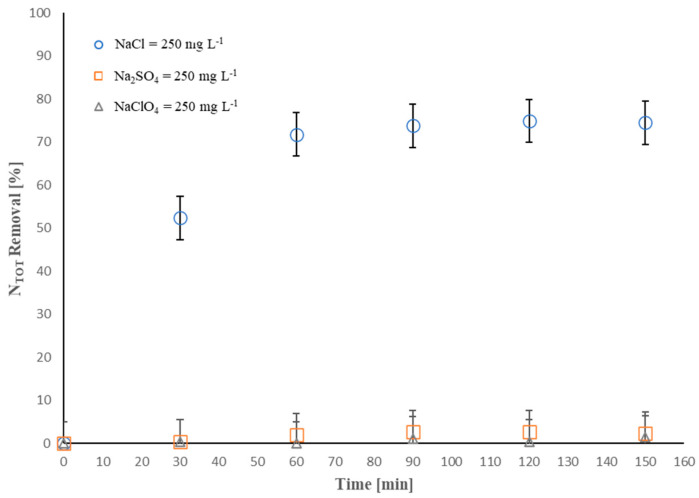
Effect of electrolyte composition on *N_TOT_* removal (%): I = 0.15 A, [*N_TOT_*]_0_ = 13 mg·L^−1^; anode active area = 50 cm^2^.

**Figure 2 molecules-28-01306-f002:**
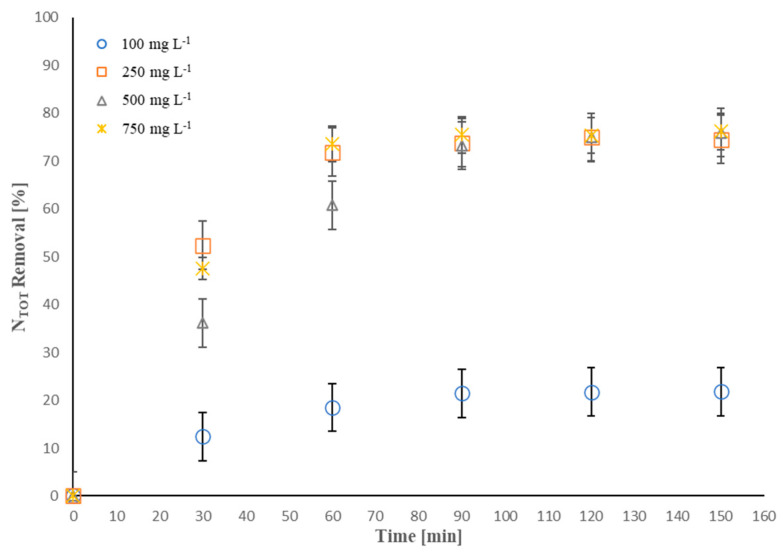
Effect of chloride concentration on *N_TOT_* removal (%): I = 0.15 A, electrolyte composition = NaCl, [*N_TOT_*]_0_ = 13 mg·L^−1^; anode active area = 50 cm^2^.

**Figure 3 molecules-28-01306-f003:**
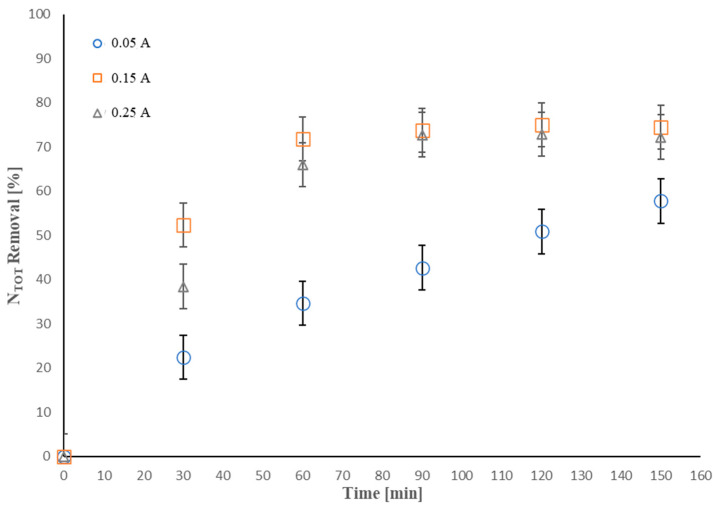
Effect of current intensity on *N_TOT_* removal (%): I = 0.15 A, electrolyte composition = NaCl, electrolyte concentration = 250 mg·L^−1^, [*N_TOT_*]_0_ = 13 mg·L^−1^, anode active area = 50 cm^2^.

**Figure 4 molecules-28-01306-f004:**
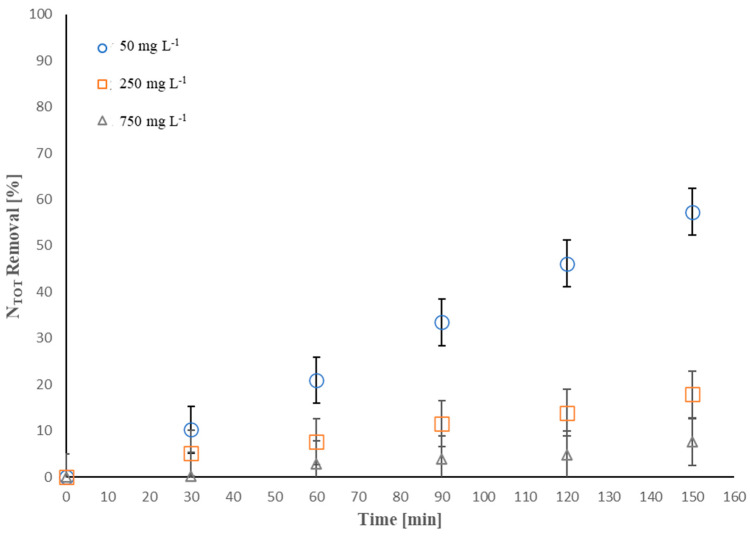
Effect of the initial ammonium concentration on *N_TOT_* removal (%): I = 0.15 A, electrolyte composition = NaCl, electrolyte concentration = 250 mg·L^−1^ [*N_TOT_*]_0_ = 13–200 mg·L^−1^, anode active area = 50 cm^2^.

**Figure 5 molecules-28-01306-f005:**
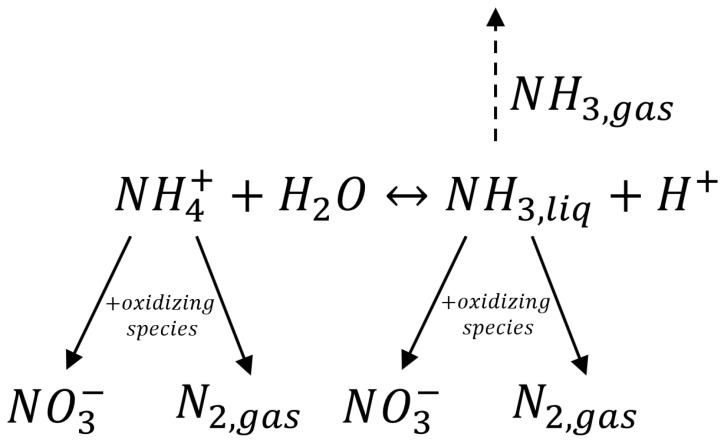
Simplified schematization of the nitrogen electrochemical oxidation mechanism.

**Figure 6 molecules-28-01306-f006:**
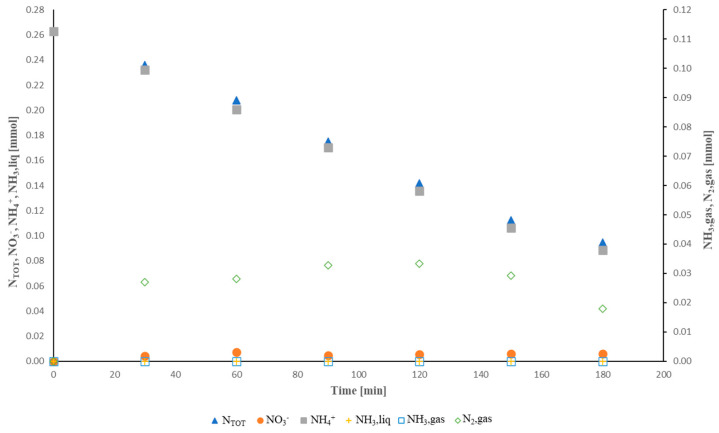
Nitrogen mass balance.

**Figure 7 molecules-28-01306-f007:**
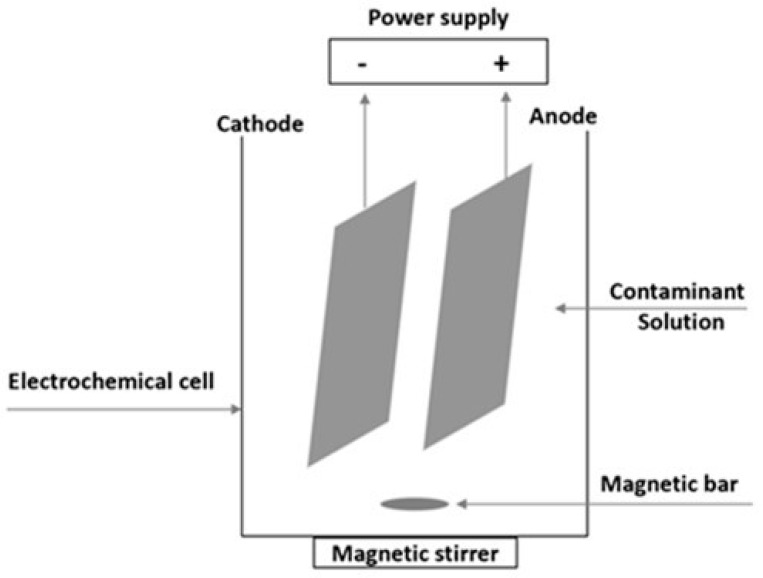
Schematization of the electrochemical reactor.

**Table 1 molecules-28-01306-t001:** Experimental conditions.

Exp. Run	Operating Conditions (*N_TOT_*_(0)_ ≈ 13–200 mg·L^−1^; V = 0.250 L; T = 25 °C; Treatment Time = 0–180 min)	Parameter Varied
1	Electrolyte concentration = 250 mg·L^−1^ M; I = 0.15 A; *N_TOT_* source = NH_4_Cl	Electrolyte type = NaCl, Na_2_SO_4_, NaClO_4_
2	Electrolyte type = NaCl; I = 0.15 A; *N_TOT_* source = NH_4_Cl	Electrolyte concentration = 100–750 mg·L^−1^
3	Electrolyte type = NaCl; electrolyte concentration = 250 mg·L^−1^; *N_TOT_* source = NH_4_Cl	I = 0.05 − 0.25 A
4	Electrolyte type = NaCl; electrolyte concentration = 250 mg·L^−1^; I = 0.15 A	*N_TOT_* source = NH_4_Cl

**Table 2 molecules-28-01306-t002:** Parameters used for the assessment of the volumetric mass transfer coefficient and the nitrogen balance.

Parameter	Value	Unit
*d_b_*	1	mm
*D_gas,liquid_*	1.5 × 10^−9^	m^2^·s^−1^
*μ_l_*	0.000825	Pa·s^−1^
*ρ_l_*	997.05	kg·m^−3^
*σ_l_*	72.80	N·m^−1^
*g*	9.81	m·s^−2^
*K_H,NH_* * _3_ *	1.7	m^3^·Pa·mol^−1^

## Data Availability

The data are available on request from both corresponding authors.

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
