# Peer review of "Electrochemical Removal of Nitrogen Compounds from a Simulated Saline Wastewater"

_molecules, 2023, doi:10.3390/molecules28031306_

Round 1

Reviewer 1 Report

Manuscript Number: molecules-2139784

Title: Electrochemical removal of nitrogen compounds from a simulated saline wastewater

This paper reported on the investigation of electrochemical removal of nitrogen compounds from synthetic saline wastewater through a lab-scale experimental reactor. In general, the topic is of interest to the wastewater treatment technology. Overall, the goal of the present paper is good, and its can be considered for the Molecules. However, several sections and contents for this current manuscript require an improvements and authors should be paying special attention to the following comments;

 1.       Abstract: Abstract should be written concise and precisely. Both quantitative and qualitative analysis should be highlighted. For instance optimum operating conditions and percentage of NTOT removal should be highlighted.

 2.       Material and methods: Section 2.3; It is suggested to provide used properties value in table form.

 3.       Results & discussion: Figure 2; the authors should explain the profile for NaCl. What happen occur at time below 60 min operation which shows the NTOT removal increase exponentially before constant with time. Please clarify.

 4.       Results & discussion: Figure 4; What is the active area size of the cathode and anode? How the applied current effect on the removal if different electrode size is used? If the size significant reflects on the performance, it is suggested to provide the electrode size or the amount of applied should be stated in current density. Please clarify.

 5.       Results & discussion: Figure 4; Please discuss the profile of each applied current. For instance, why at low applied current, the NTOT removal has still increase after 150 min compare at higher applied current which show a constant trend?

 6.       Conclusion: Conclusion should be written concise and precisely. Both quantitative and qualitative analysis should be highlighted.

Author Response

Reviewer’s general comment

This paper reported on the investigation of electrochemical removal of nitrogen compounds from synthetic saline wastewater through a lab-scale experimental reactor. In general, the topic is of interest to the wastewater treatment technology. Overall, the goal of the present paper is good, and its can be considered for the Molecules. However, several sections and contents for this current manuscript require an improvements and authors should be paying special attention to the following comments.

Authors’ response

The Authors wish to thank the Reviewer for their productive comments, helpful words and the time spent revising the manuscript. All the Reviewer’s suggestions were considered, and some corrections were made to improve the manuscript.

1) Reviewer’s comments and suggestions for Authors

Abstract: Abstract should be written concise and precisely. Both quantitative and qualitative analysis should be highlighted. For instance optimum operating conditions and percentage of NTOT removal should be highlighted.

Authors’ response

The Authors agree with the comment and wish to thank the Reviewer for pointing this out. The abstract has been modified. The changes are tracked into the revised version of the manuscript using MS Word Tracking Mode.

2) Reviewer’s comments and suggestions for Authors

Material and methods: Section 2.3; It is suggested to provide used properties value in table form.

Authors’ response

The Authors wish to thank the Reviewer for their helpful comments. The table has been added to section 2.4. Degradation mechanism and nitrogen molar balance of the revised version of the manuscript. The change is tracked using MS Word Tracking Mode.

3) Reviewer’s comments and suggestions for Authors

Results & discussion: Figure 2; the authors should explain the profile for NaCl. What happen occur at time below 60 min operation which shows the NTOT removal increase exponentially before constant with time. Please clarify.

Authors’ response

The Authors would like to thank the Reviewer for their valuable suggestion. As shown in Fig. 2, when NaCl was applied as an electrolyte, NTOT removal increased exponentially until 60 min of treatment. As reported in the literature, over time of treatment, the reactive chlorine species present in the solution can lead to the formation of undesired by-products, such as chlorate and perchlorate, among others, which may hinder further oxidation of the contaminant at the BDD anode. Consequently, after 60 min of treatment, a stable NTOT removal evolution occurred. This trend is consistent with previous electrochemical oxidation studies.

The following paragraph was added in section 3.1. Electrolyte composition impact on the electroremoval of ammonium chloride. The change is tracked using MS Word Tracking Mode.

3.1. Electrolyte composition impact on the electroremoval of ammonium chloride, lines 252-259 (revised version):

In particular, NTOT removal rapidly increased until 60 min of treatment: after that, it was constant. This trend is consistent with previous electrochemical oxidation investigations [63] and can be explained by considering that over time of treatment, the reactive chlorine species present in the solution can lead to the formation of undesired by-products, such as chlorate and perchlorate, among others, which may hinder further oxidation of the contaminant at the BDD anode [64–66]. Consequently, after 60 min of treatment, a constant NTOT removal evolution occurred.

4) Reviewer’s comments and suggestions for Authors

Results & discussion: Figure 4; What is the active area size of the cathode and anode? How the applied current effect on the removal if different electrode size is used? If the size significant reflects on the performance, it is suggested to provide the electrode size or the amount of applied should be stated in current density. Please clarify.

Authors’ response

The Authors would like to thank the Reviewer for their helpful recommendation. The active area size of the electrodes was 50 cm2 for each one; this data has been specified in section 2.2. Electrochemical oxidation experiments and in the caption of Figures 2-5. Theoretically, the larger the immersed anode area, the higher the voltage necessary for the system to act at constant current intensity. It is worth highlighting that Gorte et al. stated[1]: “For the electrochemical oxidation in the anode, increasing surface area will only help if one can ensure that there is good ionic connectivity between the electrolyte and the active site”.

5) Reviewer’s comments and suggestions for Authors

Results & discussion: Figure 4; Please discuss the profile of each applied current. For instance, why at low applied current, the NTOT removal has still increase after 150 min compare at higher applied current which show a constant trend?

Authors’ response

The Authors would like to thank the Reviewer for their valuable suggestion. Acting at the lower applied current intensity, the removal of ammoniacal nitrogen is slower. However, Piya-areetham et al.[2] reported that operating at low current intensity, the contaminant removal still increased after 360 min of treatment, confirming the trend shown in our work. Increasing the current intensity from 0.05 A to 0.15 A resulted in increasing the NTOT removal rate; this achievement can be explained by considering a faster production of hydroxyl radicals and reactive chlorine species, as stated in several previous studies on electrochemical oxidation processes. Similar trends were shown in several previous studies. 

The discussion regarding Figure 4 has been improved. The changes are reported below and tracked into the revised version of the manuscript by using MS Tracking Mode.

3.2.2. Effect of applied current intensity, lines 314-324 (original version):

NTOT removal efficiency of 57.8 % was obtained operating with the lowest applied current intensity of 0.05 A, after 150 min of treatment. As can be seen, a rise of the current intensity to 0.15 A resulted in an improvement of NTOT removal of 74.5 %, but a further increase to 0.25 A has led to basically no enhancement of the efficiency, achieving a NTOT removal of 72.3 %, at the end of the treatment. In theory, increasing the applied current intensity entails a consequent higher production of active chlorine species, speeding up the oxidation reaction and enhancing the process removal efficiencies [66,67].

However, it also means a decrease in both selectivity and current efficiency of the system since a higher applied current intensity implies an increase of undesired side reactions, such as oxygen evolution and generation of by-products (chlorate and perchlorate) [35,68]: 

WAS CHANGED TO

3.2.2. Effect of applied current intensity, lines 361-388 (revised version):

NTOT removal efficiency of 57.8 % was obtained operating with the lowest applied current intensity of 0.05 A after 150 min of treatment, despite a linear increase observed. A rise of the current intensity to 0.15 A resulted in an improvement of NTOT removal of 74.5 % after a treatment time of 90 min; after that, the removal efficiency was constant. A further increase to 0.25 A has led to no enhancement of the efficiency, showing a reduction of the removal efficiency at about 72% after 150 min.

Acting at the lower applied current intensity, the removal of ammoniacal nitrogen is slower [53]. However, Piya-areetham et al. [77] reported that operating at lower current in-tensity, the contaminant removal still increased after 360 min of treatment, confirming the trend shown in our work.

Increasing the current intensity from 0.05 A to 0.15 A resulted in increasing the NTOT removal rate. This achievement can be explained by considering a faster production of hydroxyl radicals and reactive chlorine species, as stated in several previous studies on electrochemical oxidation processes [78], which favour the oxidation of the organic compounds, i.e. the achievement of higher removal efficiencies. On the other hand, over time of treatment, the reactive chlorine species present in the solution can lead to the formation of undesired by-products, such as chlorate and perchlorate, among others, which may hinder further oxidation of the contaminant at the BDD anode [64–66], justifying the constant removal efficiency after 90 min. Similar trends were shown in several previous studies [53,79].

In theory, increasing the applied current intensity implies a consequent higher pro-duction of active chlorine species, speeding up the oxidation reaction and enhancing the process removal efficiencies [63,80]. However, it also means a decrease in both selectivity and current efficiency of the system since a higher applied current intensity implies an in-crease of undesired side reactions, such as oxygen evolution and generation of by-products (chlorate and perchlorate) [38,81], which reduces the removal efficiency:

6) Reviewer’s comments and suggestions for Authors

Conclusion: Conclusion should be written concise and precisely. Both quantitative and qualitative analysis should be highlighted.

Authors’ response

The Authors agree with the comment and wish to thank the Reviewer for pointing this out. The abstract has been modified. The changes are tracked into the revised version of the manuscript using MS Word Tracking Mode.

[1] R.J. Gorte , J.M. Gorte;  Novel SOFC anodes for the direct electrochemical oxidation of hydrocarbons, Journal of Catalysis, 2003.

[2] Reference 77 in the revised version of the manuscript.

Reviewer 2 Report

The study here by Iovino et al describes a method for electrochemical removal of nitrogen from simulated saline wastewater. As mentioned by the authors, there are several reports which already talk about similar methodology and this study does not seem a giant leap over those existing reports. The studies and experiments are adequate yet they do not seem like a big contribution towards addressing the goal of the paper as a step in advance to the previously available reports by others. Therefore this manuscript cannot be accepted for publication at this stage. However, if the authors can clearly provide a compelling argument as to why and how their findings are superior to the previous literature, there may be a chance for acceptance in future - but not at this stage. 

Author Response

Reviewer’s general comment

The study here by Iovino et al describes a method for electrochemical removal of nitrogen from simulated saline wastewater. As mentioned by the authors, there are several reports which already talk about similar methodology and this study does not seem a giant leap over those existing reports. The studies and experiments are adequate yet they do not seem like a big contribution towards addressing the goal of the paper as a step in advance to the previously available reports by others. Therefore this manuscript cannot be accepted for publication at this stage. However, if the authors can clearly provide a compelling argument as to why and how their findings are superior to the previous literature, there may be a chance for acceptance in future - but not at this stage. 

Authors’ response

The Authors wish to thank the Reviewer for the time spent on revising the manuscript. With this work the Authors aim at contributing to the exploration of nitrogen removal via electrochemical oxidation, which is a field not deeply explored, in contrast to the removal of other compounds, and still debated. The manuscript proposes an integrated approach for the assessment of the effectiveness of the nitrogen electrochemical removal, i.e. if the nitrogen was oxidized at N2, which represent the optimal result, or at other species such as NO3-. Therefore, the by-product formation was investigated, and a nitrogen balance was specified, which required the definition of a degradation mechanism and the assessment of the volumetric mass transfer coefficient. Moreover, the effects of the variation of several operative conditions on the nitrogen removal efficiency was analysed. The approach proposed resulted to be effective to investigate the main nitrogen species formed and the tests by varying the operative conditions highlighted the requirement to optimize the process; for example, it is not necessary to increase the current intensity to increase the removal efficiency. 

Reviewer 3 Report

All the comments are attached here with word document.

Author Response

Reviewer’s general comment

In general, this is a solid work generated some interesting and reliable results. The author focused on removal of nitrogen compounds by electrochemical means. However, the report of this work has some significant drawbacks, and some claims are even misleading, I suggest accepting this work after major correction. In addition, some suggestions basing on the insufficient justification and analysis of data in this present work are also given, as shown in follows:

Authors’ response

The Authors wish to thank the Reviewer for their constructive comments, for the encouraging words and for the time spent on revising the manuscript. All the Reviewer’s suggestions were considered, and some corrections were made to improve the manuscript.

1) Reviewer’s comments and suggestions for Authors

Given the topic and scope of the paper, some recently MFC work (Frontiers in Nanotechnology, 4 (2022) 1–16; Chemical Engineering Journal Advances, 10 (2022) 100283; Sensing and Bio-Sensing Research, 36 (2022) 100484; Journal of Chemical Reviews, 2021 3(4) 320–344; Journal of Nanomaterials, 2021 (2021) 1–21; Nano-Structures & Nano-Objects, 25 (2021) 100663; All Life, 14 (2021) 541–568; should be highlighted in the introduction and even in the discussion part to broaden the readership.

Authors’ response

The Authors would like to thank the Reviewer for their helpful remark, and the useful references suggested, which were included in the manuscript. The changes are tracked into the revised version of the manuscript using MS Word Tracking Mode.

2) Reviewer’s comments and suggestions for Authors

Please carefully check the sentences again. I strongly encourage the authors to ask a native English speaker to brush up English.

Authors’ response

The Authors would like to thank the Reviewer for their useful recommendation. The sentences have been carefully checked. The modifications are tracked into the manuscript by using MS Word Tracking Mode.

3) Reviewer’s comments and suggestions for Authors

What can you say about the novelty of this work?

Authors’ response

With this work, the Authors aim to contribute to the exploration of nitrogen removal via electrochemical oxidation, which is a field not deeply explored, in contrast to the removal of other compounds and still debated. The manuscript proposes an integrated approach for assessing the effectiveness of the electrochemical nitrogen removal, i.e. if the nitrogen was oxidized at N2, representing the optimal result, or at other species such as NO3-. Therefore, the by-product formation was investigated, and a nitrogen balance was specified, which required the definition of a degradation mechanism and the assessment of the volumetric mass transfer coefficient. Moreover, the effects of the variation of several operative conditions on the nitrogen removal efficiency were analysed. The approach proposed resulted in being effective in investigating the main nitrogen species formed, and the tests by varying the operative conditions highlighted the requirement to optimize the process; for example, it is not necessary to increase the current intensity to increase the removal efficiency.

The introduction was modified by adding the following paragraph. The change is tracked by using MS Word Tracking Mode.

Introduction, lines 124-131 (revised version)

With this work, we aim to contribute to the exploration of nitrogen removal via electrochemical oxidation, a field not deeply explored in contrast to the removal of other compounds and still debated. The manuscript proposes an integrated approach for assessing the effectiveness of the electrochemical nitrogen removal, i.e. if the nitrogen was oxidized at N2, representing the optimal result, or at other species such as NO3-. Therefore, the by-product formation was investigated, and a nitrogen balance was performed, which required the definition of a degradation mechanism and the assessment of the volumetric mass transfer coefficient.

4) Reviewer’s comments and suggestions for Authors

Why direct electrochemical oxidation resulted to be ineffective for nitrogen compound removals? Briefly discuss with reference?

Authors’ response

The Authors wish to thank the Reviewer for the interesting observation. Candido et al.[1] reported that the direct electrochemical oxidation of ammonium nitrogen can be affected by incompletely formed oxidized nitrogen species and OHgenerated during the process on the anode surface, resulting in a reduction of the oxidation efficiency. This achievement has also been reported in several studies, in which it is highlighted that one of the main limitations of ammoniacal nitrogen oxidation is represented by the competition between the adsorption of ammoniacal species and OH on the anode surface, causing a blocking effect on its active zones. The results showed in our work are consistent with previous studies where it is reported that nitrogen compounds are known to be the main species poisoning (deactivating) anode surface during ammoniacal nitrogen oxidation.

Some clarifications have been included in section 3.1. Electrolyte composition impact on the electroremoval of ammonium chloride. The changes are tracked by using MS Word Tracking Mode and reported below:

3.1. Electrolyte composition impact on the electroremoval of ammonium chloride, lines 285-294 (revised version):

Candido et al. [16] reported that the direct electrochemical oxidation of ammonium nitrogen can be affected by the formation of incompletely oxidized nitrogen species and •OH generated during the process on the anode surface, resulting in a reduction of the oxidation efficiency. This achievement has also been reported in several studies, in which it is high-lighted that one of the main limitations of ammoniacal nitrogen oxidation is represented by the competition between the adsorption of ammoniacal species and •OH on the anode surface, causing a blocking effect on its active zones [68,69]. The results showed in our work are consistent with previous studies [64], where it is reported that nitrogen compounds are known to be the main species poisoning (deactivating) anode surface during ammoniacal nitrogen oxidation.

5) Reviewer’s comments and suggestions for Authors

Please give attentions on writing subscripts, superscript, and radical writings.

Authors’ response

The Authors would like to thank the Reviewer for their useful comment. Both the text and the figures were checked, and writing subscripts, superscripts, and radical writings were fixed.

6) Reviewer’s comments and suggestions for Authors

Please revise the detailed degradation mechanism of such compounds?

Authors’ response

The Authors would like to thank the Reviewer for their valuable advice. The degradation mechanism proposed in the study is consistent with previous works. Yao et al.[2] showed that the mechanism of ammoniacal nitrogen removal depends on both the hydroxyl radical and active chlorine, suggesting that the contaminant could be efficiently oxidized by these oxidants. Several works reported that nitrogen gas and nitrate are the main products of electrochemical oxidation of ammoniacal nitrogen.

Some clarifications have been included in section 3.3. Degradation mechanism and nitrogen molar balance. The changes are tracked by using MS Word Tracking Mode and reported below:

3.3. Degradation mechanism and nitrogen molar balance, lines 425-429 (revised version):

Yao et al. [53] showed that the mechanism of ammoniacal nitrogen removal depends on both the hydroxyl radical and active chlorine, suggesting that the contaminant could be efficiently oxidized by these oxidants. Several works reported that nitrogen gas and nitrate are the main products of electrochemical oxidation of ammoniacal nitrogen [35,64,84].

7) Reviewer’s comments and suggestions for Authors

Please try to write the numerical temperature instead of room temperature.

Authors’ response

The Authors would like to thank the Reviewer for the useful recommendation. The numerical temperature instead of “room temperature” has been specified (25°C). The change is tracked in the revised version of the manuscript by using MS Word Tracking Mode.

8) Reviewer’s comments and suggestions for Authors

During your electrochemical system, why you prefer two electrode system? Why not three electrode system? Write your reason with possible citation.

Authors’ response

The Authors would like to thank the Reviewer for their interesting comment. The system used in our study, consisting of a batch reactor with 2 vertical parallel electrodes, is the most employed configuration at lab-scale for removing ammoniacal nitrogen and other contaminants due to its easiness of installation, handling, sampling and efficiency.

Some clarifications have been included in section 3.3. Degradation mechanism and nitrogen molar balance. The changes are tracked by using MS Word Tracking Mode and reported below:

2.2. Electrochemical oxidation experiments, lines 160-162 (revised version):

The reactor configuration used in this study is the most employed one at the lab-scale for removing ammoniacal nitrogen and other contaminants due to its easiness of installation, handling, sampling and efficiency [51–54].

9) Reviewer’s comments and suggestions for Authors

What was the separation distance between Anode and Cathode electrodes?

Authors’ response

The gap between the anode and cathode was 1 cm. The data is specified in the section 2.2. Electrochemical oxidation experiments.

10) Reviewer’s comments and suggestions for Authors

Your electrochemical system is non-spontaneous (i.e, the system or cell directly use a direct power supply). Hence, did you think that this cell (reactor) is cost-effective for large scale applications?

Authors’ response

The Authors wish to thank the Reviewer for the interesting consideration. Electro-oxidation processes are considered energy-consuming systems, so the electric energy contribution is the most significant parameter to consider for estimating total operational costs. Anyway, total operating costs also imply electrode supply and replacement, pumping, stirring, cleaning, and maintenance procedures[3]. Even though batch configurations result in being highly effective for removing contaminants from impacted water3,4, the operation costs can be decreased by increasing the scale of the EO process and acting in continuous configurations5-7. Anyway, in future work, the Authors will deeply investigate the cost-effectiveness of the process at a large scale. 

11) Reviewer’s comments and suggestions for Authors

In most electrochemical applications, using KCl as electrolyte is more preferable than NaCl. Therefore, why you choose NaCl?

Authors’ response

The Authors wish to thank the Reviewer for the interesting comment. Parsa et al.8 in their electrochemical oxidation study reported that NaCl was more effective as a supporting electrolyte than KCl, for removing azo dyes in aqueous media. Many previous studies used NaCl as a supporting electrolyte to remove ammoniacal nitrogen and other contaminants from water by electrochemical oxidation3,9-12.

12) Reviewer’s comments and suggestions for Authors

Line …184 and 185.. the statement says “where Sc and Re are Schmidt and Reynolds numbers, calculated according to the following equations, respectively (10.1036/0071422943):” from this replace DOI number “10.1036/0071422943” by the exact reference citation.

Authors’ response

The Authors wish to thank the Reviewer for the useful comment and apologize for the mistake. DOI number has been replaced by the exact reference citation.

13) Reviewer’s comments and suggestions for Authors

In figure 4, it is known that at high applied current nitrogen removal efficiency is expected to be maximum. But from your graph removal percentage is decreases when time is going on. So, could you suggest this reason in brief?

Authors’ response

The Authors wish to thank the Reviewer for the interesting observation. Theoretically, by increasing the applied current intensity, the process removal efficiency should be enhanced. However, it also means a decrease in both the selectivity and current efficiency of the system since a higher applied current intensity implies an increase of undesired side reactions, which reduces the removal efficiency. 

Some clarifications have been included in section 3.2.2. Effect of applied current intensity. The changes are tracked by using MS Word Tracking Mode.

14) Reviewer’s comments and suggestions for Authors

Conclusion part must be revised and some future recommendations from your findings must be included.

Authors’ response

The Authors agree with the comment and wish to thank the Reviewer for pointing this out. The abstract has been modified. The changes are tracked into the revised version of the manuscript using MS Word Tracking Mode.

[1] Reference 16 in the revised version of the manuscript.

[2] Reference 53 in the revised version of the manuscript.

[3] Kaur et. al, Evaluation and disposability study of actual textile wastewater treatment by electro-oxidation method using Ti/RuO2 anode, Process Safety and Environmental Protection, 2017.

3 Yao et al. Process Optimization of Electrochemical Oxidation of Ammonia to Nitrogen for Actual Dyeing Wastewater Treatment, International Journal of Environmental Research and Public Health, 2019.

4 Solano et al. Decontamination of real textile industrial effluent by strong oxidant species electrogenerated on diamond electrode: Viability and disadvantages of this electrochemical technology, Applied Catalysis B: Environmental, 2013.

5 Kumar et al. Electro-oxidation of nitrophenol by ruthenium oxide coated titanium electrode: Parametric, kinetic and mechanistic study, Chemical Engineering Journal, 2015.

6 Fenti et al. Performance testing of mesh anodes for in situ electrochemical oxidation of PFAS, Chemical Engineering Journal Advances, 2021.

7 Chatzisymeon et al. Electrochemical treatment of textile dyes and dyehouse effluents, Journal of Hazardous Materials, 2006.

8 Parsa et al. Electrochemical oxidation of an azo dye in aqueous media investigation of operational parameters and kinetics, Journal of Hazardous Materials, 2009.

9 Li et al. Ammonia removal in electrochemical oxidation: Mechanism and pseudo-kinetics, Journal of Hazardous Materials, 2009.

10 Kapalka et al., Direct and mediated electrochemical oxidation of ammonia on boron-doped diamond electrode, Electrochemistry Communications, 2010.

11 Scialdone et al., Electrochemical oxidation of organics in water: Role of operative parameters in the absence and in the presence of NaCl, Water Research, 2009.

12 Salvestrini et al. Electro-Oxidation of Humic Acids Using Platinum Electrodes: An Experimental Approach and Kinetic Modelling, Water, 2020.

Reviewer 4 Report

The manuscript entitled "Electrochemical removal of nitrogen compounds from a simulated saline wastewater" has an interesting and novelty work. The introduction and the relevant references are sufficient. The presentation of the results is well explained and the conclusions are supported by the results. However,

In line 185:what do you mean by this number(10.1036/0071422943).

In fig 7: the y-axis should be corrected to be ( N2 ,, NH2).

Finally, the paper is acceptable taking into consideration the above corrections.

Author Response

Reviewer’s general comment

The manuscript entitled "Electrochemical removal of nitrogen compounds from a simulated saline wastewater" has an interesting and novelty work. The introduction and the relevant references are sufficient. The presentation of the results is well explained and the conclusions are supported by the results.

Authors’ response

The Authors wish to thank the Reviewer for their positive and constructive words and the time spent revising the manuscript. All the Reviewer’s suggestions were considered, and some corrections were made to improve the manuscript.

1) Reviewer’s comments and suggestions for Authors

In line 185:what do you mean by this number (10.1036/0071422943).

Authors’ response

The Authors wish to thank the Reviewer for the helpful comment and apologize for the mistake. DOI number has been replaced by the exact reference citation.

2) Reviewer’s comments and suggestions for Authors

In fig 7: the y-axis should be corrected to be ( N2 ,, NH2).

Authors’ response

The Authors wish to thank the Reviewer for the helpful suggestion. Figure 7 has been modified and included in the revised version of the manuscript.

Round 2

Reviewer 2 Report

The authors have provided a synopsis of their work as a part of their response. However, following my previous comment "if the authors can clearly provide a compelling argument as to why and how their findings are superior to the previous literature, there may be a chance for acceptance in future - but not at this stage. ", it appears that this response is still lacking. Electrochemical approaches are less explored but the authors should comment as to what additional improvements does this work provide? How is this work helpful to extend the horizon for this field? What are the areas they think they have touched which none of the other works have ? This does not mean a summary of the paper but a scientific evaluation of the merit of this work. I would like to provide the authors one more chance to submit an improved introduction before it is accepted for publication. 

Author Response

Reviewer’s general comment

The authors have provided a synopsis of their work as a part of their response. However, following my previous comment "if the authors can clearly provide a compelling argument as to why and how their findings are superior to the previous literature, there may be a chance for acceptance in future - but not at this stage. ", it appears that this response is still lacking. Electrochemical approaches are less explored but the authors should comment as to what additional improvements does this work provide? How is this work helpful to extend the horizon for this field? What are the areas they think they have touched which none of the other works have ? This does not mean a summary of the paper but a scientific evaluation of the merit of this work. I would like to provide the authors one more chance to submit an improved introduction before it is accepted for publication.

Authors’ response

The Authors wish to thank the Reviewer for the additional time dedicated to revising the manuscript and for the relevant suggestions.

Ammoniacal nitrogen removal via electrochemical oxidation is a field debated as the effects of electrolyte composition, chloride concentration, and current intensity on removal efficiency are controversial. Therefore, with this work investigating the parameters mentioned above, the Authors would like to contribute to stepping forward towards understanding and explaining the process. Moreover, the Authors’ contribution has also been related to defining an integrated approach based on compound measurements, definition of a degradation mechanism, mass transfer assessment and material balance performance. According to the best of the Authors’ knowledge, this is the first time this kind of approach has been proposed for the investigation of ammoniacal nitrogen removal from wastewater. Thanks to this approach, it is possible to assess the effectiveness of the electrochemical nitrogen removal, i.e. if the nitrogen was oxidized at N2, representing the optimal result, or at other species such as NO3-.

Some clarifications have been included in section 1. Introduction. The changes are tracked by using MS Word Tracking Mode and reported below:

Introduction, lines 116-137 (revised version)

With this work, we aim to contribute to the exploration of nitrogen removal via electrochemical oxidation, a field not deeply explored in contrast to the removal of other compounds and still debated. The effects of electrolyte composition, chloride concentration, and current intensity on removal efficiency are controversial. For example, Kapalka et al. [51] indicated that the direct EO pathway could oxidize ammonia on the BDD anode surface. This result was confirmed by Zollig et al. [52]. Conversely, Candido et al. [16] reported a poor contribution of direct EO on the removal of ammoniacal nitrogen, likely due to the possible formation of incompletely oxidized adsorbed nitrogen species and OH on the anode surface, resulting in shielding for the direct oxidation. Mandal et al. [35] reported that the ammonia oxidation increased when the initial chloride concentration increased from 300 to 1500 mg L-1; but in the range from 300 to 900 mg L-1 the ammonia removal percentage did not change significantly. Unlike, Li et al. [53] showed a linear correlation between the ammonia removal efficiency and the initial chloride concentration in all the investigated range, confirming the results of other studies [54,55]. Shih et al. [56] reported an appreciable impact of the applied current intensity on the EO of ammoniacal nitrogen: the higher the current intensity, the higher the removal of the contaminant. This trend was confirmed by Zhang et al.[55], but contradicted other previous studies where a decrease in removal efficiency was found since a higher applied current intensity implies an increase of undesired side reactions, such as oxygen evolution and generation of by-products [38,81]. Therefore, by investigating the parameters mentioned above, the Authors would like to contribute to stepping forward towards understanding and explaining nitrogen removal via electrochemical oxidation.

Moreover, the manuscript proposes an integrated approach for assessing the effectiveness of the electrochemical nitrogen removal, i.e. if the nitrogen was oxidized at N2, representing the optimal result, or at other species such as NO3-. Therefore, the by-product formation was investigated and monitored, and a nitrogen balance was performed, which required the definition of a degradation mechanism and the assessment of the volumetric mass transfer coefficient. According to the best of the Authors’ knowledge, this is the first time this kind of integrated approach has been proposed for the investigation of ammoniacal nitrogen removal from wastewater.

Reviewer 3 Report

All comments are revised and thus the paper be considered for publication in the present form. 

Author Response

Reviewer’s general comment

All comments are revised and thus the paper be considered for publication in the present form.

Authors’ response

The Authors wish to thank the Reviewer for their constructive comments and suggestions, thanks to which the paper has been improved.
